# Peripheral Nerve Injury Induces Changes in the Activity of Inhibitory Interneurons as Visualized in Transgenic GAD1-GCaMP6s Rats

**DOI:** 10.3390/bios12060383

**Published:** 2022-06-01

**Authors:** Vijai Krishnan, Lauren C. Wade-Kleyn, Ron R. Israeli, Galit Pelled

**Affiliations:** 1Department of Mechanical Engineering, Michigan State University, East Lansing, MI 48824, USA; krish102@msu.edu; 2Neuroscience Program, Michigan State University, East Lansing, MI 48824, USA; wadelau2@msu.edu; 3Department of Biomedical Engineering, Michigan State University, East Lansing, MI 48824, USA; israeli2@msu.edu; 4Department of Radiology, Michigan State University, East Lansing, MI 48824, USA

**Keywords:** pain, somatosensory cortex, calcium imaging, corpus callosum, plasticity, rehabilitation

## Abstract

Peripheral nerve injury induces cortical remapping that can lead to sensory complications. There is evidence that inhibitory interneurons play a role in this process, but the exact mechanism remains unclear. Glutamate decarboxylase-1 (GAD1) is a protein expressed exclusively in inhibitory interneurons. Transgenic rats encoding GAD1–GCaMP were generated to visualize the activity in GAD1 neurons through genetically encoded calcium indicators (GCaMP6s) in the somatosensory cortex. Forepaw denervation was performed in adult rats, and fluorescent Ca^2+^ imaging on cortical slices was obtained. Local, intrahemispheric stimulation (cortical layers 2/3 and 5) induced a significantly higher fluorescence change of GAD1-expressing neurons, and a significantly higher number of neurons were responsive to stimulation in the denervated rats compared to control rats. However, remote, interhemispheric stimulation of the corpus callosum induced a significantly lower fluorescence change of GAD1-expressing neurons, and significantly fewer neurons were deemed responsive to stimulation within layer 5 in denervated rats compared to control rats. These results suggest that injury impacts interhemispheric communication, leading to an overall decrease in the activity of inhibitory interneurons in layer 5. Overall, our results provide direct evidence that inhibitory interneuron activity in the deprived S1 is altered after injury, a phenomenon likely to affect sensory processing.

## 1. Introduction

Peripheral nerve injury (PNI) is characterized by abnormal pathologies in sensory and motor pathways. It is often accompanied by neuropathic and phantom limb pain, leading to poor prognosis and recovery. Substantial research shows that PNI and sensory deprivation prompt a complex sequence of changes in neural activity that lead to the remapping of cortical representations in humans [1,2], non-human primates [3], and rodent brains [4]. Evidence suggests that this plasticity dictates the degree of sensory complications [5,6]. Therefore, identifying the neural circuits and the plasticity mechanism associated with PNI is essential in developing new and improved treatment strategies to minimize post-injury complications.

Removing peripheral input affects multi–synaptic pathways, including thalamocortical connections, intrahemispheric connections, and interhemispheric connections. Strengthening of thalamocortical synapses after PNI in rats has been documented using functional magnetic resonance imaging (fMRI) and electrophysiology [7]. Plasticity of local circuits after limb and whisker denervation has been shown to occur in the primary somatosensory cortex (S1) contralateral and ipsilateral to the side of denervation [8,9,10,11,12,13,14,15]. These studies demonstrate that both excitatory neurons and inhibitory interneurons within the deprived S1 (contralateral to the injury) are affected by the loss of input. Inhibitory interneurons are known to shape sensory integration, cortical maps, and sensory processing of stimuli [16]. Nevertheless, it remains unclear how inhibitory interneurons are affected by injury and subsequently lead to abnormal sensory processing.

Several studies suggest that injury leads to upregulation in the activity of inhibitory interneurons [9,15,17]. Possible mechanisms include the decreased activity of excitatory neurons due to a lack of thalamic input to cortical layer 4 (L4) and abnormal interhemispheric, transcallosal communication. Indeed, modulating interhemispheric communication by optogenetics decreased inhibitory activity in the deprived S1 and restored the excitation-inhibition balance [15]. Using non–invasive brain stimulation over the deprived S1 to increase activity has been shown to reduce pain and increase performance after injury [18]. On the other hand, there is evidence suggesting sensory deprivation increases cortical excitability through transcallosal communication, which may suggest downregulation in the activity of inhibitory interneurons [19,20]. Nonetheless, all these studies show that injury induces changes in the balance between excitation and inhibition in the S1 and changes in the communication between neurons in the S1. Together, this leads to abnormal sensory perception.

The goal of the present study was to determine the role of inhibitory interneurons in cortical remapping after injury. Inhibitory interneurons are typically smaller than excitatory neurons and account for only 20% of cortical neurons. Thus, recording and visualizing their activity using electrophysiology and microscopy is often challenging. Advances in transgenic technology now allow the genetic engineering of rats [21] and mice [22] to express genetically encoded calcium–sensitive proteins (GCaMPs) [23] under specific neural promoters [24,25].

Glutamate decarboxylase–1 (GAD1) is a protein expressed in inhibitory interneurons and is responsible for basal GABA production [26]. Transgenic Sprague Dawley rats were generated using the CRISPR/Cas9 system to encode GAD1–GCaMP6s. The CRISPR/Cas9 system is an effective tool for gene editing in various model organisms, including mice and humans [27]. This new transgenic rat allows for the visualization of neuronal activity in GAD1-inhibitory interneurons by measuring calcium changes.

To measure the activity of inhibitory interneurons in the present study, a bipolar tungsten electrode was positioned inside layers 2/3 (L2/3), layer 5 (L5), or in the corpus callosum (CC) of the deprived S1 in denervated and control GAD1–GCaMP6s rats. The effects of intrahemispheric stimulation were analyzed in L2/3 and L5 of the deprived S1, while the effect of interhemispheric stimulation of the CC was analyzed in L5 of the deprived S1. The results suggest that denervation leads to increased activity of inhibitory interneurons in response to local, intrahemispheric stimulation, whereas denervation impacts interhemispheric communication and leads to an overall decrease in the activity of inhibitory interneurons. Overall, our results provide direct evidence that the activity of inhibitory interneurons in the deprived S1 is altered after injury.

## 2. Materials and Methods

Animal experiments were approved by Michigan State University’s Institutional Animal Care and Use Committee and conducted according to the NIH Guide for the Care and Use of Laboratory Animals.

### 2.1. Generation of Transgenic GAD1-GCaMP6s Knock-In Rat

The rat GAD1 locus (ENSRNOG00000000007) was targeted using CRISPR–Cas9 genome editing and a long single-stranded oligodeoxynucleotides (lssODN) HDR donor template [28,29]. Selection of guide RNAs (gRNAs), locus analysis, construct design, and sequence analysis, and alignments were performed using the Benchling platform and MacVector software. A gRNA targeting exon 2 with a protospacer and protospacer adjacent motif (PAM) sequence 5′–CGTGGAAGATGCCATCAGCTCGG–3′ was chosen to generate a double–strand break (DSB) 2bp upstream of the translational start site (ATG).

An HDR donor construct was generated to include 5′ and 3′ homology arms flanking the GCaMP6s coding sequence (cds) and a P2A self–cleaving signal peptide, upstream and in–frame with the GAD1 coding region in exon 2. Homology arm (HA) regions were PCR amplified from Sprague Dawley rat genomic DNA with a Q5^®^ High–Fidelity DNA Polymerase (M0491, New England Biolabs, Ipswich, MA, USA) and primers O619F and O620R (Primer Table). The GCaMP6s cds were subcloned from vector pGP–CMV–GCaMP6s, a gift from Douglas Kim & GENIE Project (Addgene plasmid #40753). A GSG–P2A sequence was synthesized, and individual fragments were PCR–amplified with appropriate overlaps for assembly into a pBKSII backbone using the NEBuilder^®^ HiFi Assembly Cloning kit (E5520S, New England Biolabs).

To produce a lssODN donor template, a nickase–based method was employed using the Long ssDNA Preparation Kit (DS620, BioDynamics Laboratory Inc., Hackensack, NJ, USA). The GCaMP6s–P2A insert flanked by 375 bp 5′HA and 343bp 3′HA was amplified (O712F/O713R) and cloned into the nickase vector pLSODN–3. The resulting sequence–verified plasmid was digested with NsiI and the nickase Nb.BbvCI, and the released ssDNA was denatured, gel extracted, and purified using a Clontech NucleoSpin^®^ gel extraction kit (NC923380, Fisher Scientific, Waltham, MA, USA).

Sprague Dawley rats were purchased from Charles River Laboratory (Crl:Sprague Dawley, strain code 400). Ribonucleoprotein (RNP) complexes were prepared by hybridization of synthetic Alt-R^®^ CRISPR crRNA and tracrRNA, which were then complexed in equimolar amounts with [100 ng/µL] Alt-R^®^ S.p. Cas9 Nuclease V3 protein (Integrated DNA Technologies Inc., Coralville, IA, USA). RNP complexes were mixed with the lssODN donor template [10 ng/µL] and delivered into Sprague Dawley rat zygotes by pronuclear microinjection. Microinjected embryos were implanted into pseudo-pregnant recipients using standard approaches.

Founder litters were screened for correct HDR events by PCR with 5′ (O663F/O664R; O753F/756R) and 3′ (O665F/O666R; O757F/O752R) external primers. Founder T1641 was identified as having the correct insertion, and the entire cassette and surrounding genomic regions were amplified, cloned, and verified by Sanger sequencing. One histidine residue was deleted from the His-tag at the N–terminus of the GCaMP6s cds, and the remaining insert sequence and flanking genomic regions were intact. 

GAD1–GCaMP6s rats were kept heterozygous and were bred to wild–type Sprague Dawley animals for multiple generations to out–cross any potential off–target mutations. Analysis of the gRNA used for targeting with CRISPR and Benchling prediction algorithms did not identify any significant off–target hits either in exons (all CFD specificity scores <0.27) or on the same chromosome (all CFD specificity scores <0.21).

### 2.2. Peripheral Nerve Injury

Sprague Dawley adult rats (100–130 g, 5 weeks old, *n* = 12, (9 male, 3 female)) underwent forepaw denervation by excision of the radial, median, and ulnar nerves [10]. Forepaw denervation was performed by cutting the median nerve below the level of the triceps muscles and cutting the radial and ulnar nerves beneath the area of the bicep muscles. Rodents were under 2% isoflurane anesthesia while denervation was performed. As a result, both sensory and motor fiber pathways were completely severed. The incision was cleaned and closed using silk sutures and tissue glue. Tramadol (0.1 mg/300 mg) was administered orally for 5 days after the injury. For sham controls, rats underwent the entire procedure, including exposure of the nerves, followed by suturing of the skin.

### 2.3. Immunochemistry of Brain Slices

Rats were transcardially perfused with 0.1 M phosphate buffer saline solution (PBS) at pH 7.4. This was followed by an ice-cold 4% paraformaldehyde solution, and the brains were subsequently removed. Brains were cryoprotected in 30% sucrose overnight. The brain tissue was then embedded in OCT compound (Tissue–Tek) and sliced on a cryostat (Leica Microsystems GmbH, Wetzlar, Germany) to obtain 20 μm thick coronal sections, which were collected on glass slides. Sections were incubated overnight with primary antibodies to detect GAD1 (1:100; Abcam #ab97739) and GFP (1:500; Invitrogen #ab13970) at 4 °C. After incubation with the primary antibody, sections were washed with PBS (three times, 5 min each) and incubated for 3 h at room temperature with secondary antibodies (Alexa Fluor 555 & Alexa Fluor 488). Sections were washed twice with PBS, and ProLong Gold Antifade Reagent (Thermofisher Scientific, Waltham, MA, USA) on coverslips were used.

### 2.4. Confocal Imaging

Confocal images were acquired using the Nikon A1–Rsi Confocal Laser Scanning Microscope (Nikon Instruments, Inc., Tokyo, Japan) configured on a Nikon Eclipse Ti inverted microscope. Images were collected using either a Nikon 10× Plan Apo (NA 0.45) objective, a Nikon 20× Plan Apo VC (NA 0.75) objective, a Nikon 40× Plan Fluor (NA 1.30) oil objective, or a Nikon 60× Apo (NA 1.40) oil objective. Image acquisition was performed using Nikon NIS–Elements AR software (version 5.20.02). Green fluorescence was excited using a 488 nm diode laser, and fluorescence emission was detected through a 525/50 nm bandpass emission filter. Red fluorescence was excited using a 561 nm diode laser, and fluorescence emission was detected through a 595/50 nm bandpass emission filter. For each data set, a confocal series through the thickness of the tissue section was collected. For the 20× objectives, confocal images were collected in 1.5 µm increments through an average thickness of 30 µm. For the 40× objectives, confocal images were collected in 1 µm increments through an average thickness of 20 µm. For each confocal series, a Maximum Intensity Projection image was generated, representing the brightest intensity pixels through the Z–depth.

### 2.5. Calcium Imaging and Stimulation

Cortical coronal brain slices were obtained from rats 2 weeks post-PNI surgery. Rats were euthanized with isoflurane, and the brain was removed and placed in oxygenated (95% O_2_/5% CO_2_) ice–cold artificial cerebrospinal fluid (ACSF) in mM: NaCl–119, MgSO_4_·7H_2_O–1.2, KCl–2.5, NaH_2_PO_4_–1.15, Glucose–11.0, NaHCO_3_–26.2, CaCl_2_·2H_2_O–2.5. 300 µm slices were obtained using tissue vibratome (Leica Biosystems, Deer Park, IL, USA) in ice–cold ACSF. Slices were then bubbled with 95% O_2_/5% CO_2_, pH 7.4, at room temperature for 45 min before using them for experimentation. Slices were then loaded on a fixed stage microscope (DM6FS, Leica Biosystems) fitted with a Hamamatsu ORCA-fusion sCMOS camera.

Constant perfusion with ACSF was performed to ensure the physiological health of slices. GCaMP6s positive fluorescent cells in cortical L2/3 and L5 were identified and imaged with a 5x objective (1.25 internal magnification chamber, resulting in a magnification of 6.25). Identified GAD1–GCaMP6s fluorescent cell(s) were imaged as a time series experiment. Regions of interest were drawn around GAD1–GCaMP6s neurons in L2/3 and L5 using LAS X (Leica Biosystems). A bipolar tungsten electrode was positioned inside L2/3, L5, or in the CC in the deprived S1, and 100 Hz stimulation was delivered for 5 s. Fluorescence intensity changes over time were recorded with regions of interest before and after electrical stimulation in the desired cortical region.

### 2.6. Statistical Analysis

We assumed non-normality based on the Kolmogorov–Smirnov Pearson test and used the Mann–Whitney Wilcoxon test to determine significance.

## 3. Results

To visualize the activity of inhibitory interneurons, transgenic rats were generated to express GCaMP6s in GAD1+ inhibitory interneurons. Transgenic GAD1–GCaMP6s knock-in rats were generated by CRISPR–Cas9 genome editing using a long single-stranded DNA repair template. To preserve the expression of the endogenous GAD1 protein, a GCaMP6s cassette was followed by a P2A self–cleaving peptide [30] sequence and was inserted at the translational start site, in–frame with the coding sequence of GAD1.

To validate the expression of GCaMP6s and GAD1, immunohistochemistry was performed with primary antibodies against GFP and GAD1, respectively. Confocal imaging revealed GCaMP6s expression (green) throughout the cortical layers (Figure 1). GAD1 (red) expression was also observed throughout the cortical layers with a sparse labeling pattern. Examination of merged (GCaMP6s + GAD1) images revealed colocalization of GCaMP6s and GAD1 expression. Taken together, these results demonstrate the transgenic rat successfully expresses GCaMP6s in GAD1 neurons.

### 3.1. Intrahemispheric Upregulation of GAD1 Neurons in the Deprived S1

Changes in fluorescence of GCaMP6s from identified GAD1 neurons in L2/3 and L5 of the deprived S1 were collected in response to local stimulation. Representations of identified GAD1 neurons in L2/3 and L5 are demonstrated in Figure 2A,C, respectively. Experimental schematics demonstrating fluorescence intensity changes over time were recorded in L2/3 (Figure 2B) & L5 (Figure 2D) in response to electrical stimulation, 30s post basal activity. For intrahemispheric L2/3 and L5 experiments, we imaged 27 slices from denervated rats (*n* = 5) and 37 from control rats (*n* = 6). From these slices, we identified 248 GAD1–GCaMP6s positive neurons in denervated rats and 366 in control rats. The fluorescence change amplitude (Δ*A*) was calculated by taking the difference between the maximum fluorescence value after stimulation (max value from 0–10 s post-stimulation; i.e., *MaxF*) and the average fluorescence value prior to stimulation (0–29 s pre-stimulation; i.e., *BaseF*) and dividing it by *BaseF*, as represented in the following: $$\Delta A=\frac{MaxF-BaseF_{0-29s}}{BaseF_{0-29s}}$$

In denervated rats, local stimulation induced an average fluorescence change of 5.76 ± 6.49% (mean ± standard deviation (*SD*)) in L2/3 and 2.75 ± 2.77% in L5. Additionally, local stimulation in control rats induced an average fluorescence change of 0.24 ± 1.39% in L2/3 and 0.62 ± 1.65% in L5 (Figure 3A,B; Mann–Whitney Wilcoxon test, *p* < 0.0001 and *p* < 0.0001, respectively). GAD1 neurons were considered responsive to stimulation when *MaxF* was 2*SD* above *BaseF*, as indicated below:$$Responsive\geq 2SD\times BaseF_{0-29s}$$

In denervated rats, 131 out of 202 GAD1 neurons (64.85%) in L2/3 were deemed responsive, while 34 of 46 (73.91%) were deemed responsive to stimulation in L5. In control rats, significantly fewer GAD1 neurons were responsive to stimulation: 18 out of 185 (9.73%) in L2/3, and 31 of 181 (17.13%) in L5 (Figure 3C,D; L2/3 Chi–squared = 113.20, *p* < 0.0001; L5 Chi–squared = 56.88, *p* < 0.0001). Additionally, the responsive GAD1 neurons in the denervated rats had a significantly larger average amplitude change in fluorescence in L2/3 (8.64 ± 6.38%) when compared to those of the control rats (2.24 ± 1.83%; Figure 3E, Mann–Whitney Wilcoxon test, *p* < 0.0001). However, no significant difference was found for the responsive GAD1 neurons between denervated (3.45 ± 2.90%) and control rats (2.12 ± 1.56%), as shown in Figure 3F (Mann–Whitney Wilcoxon test, *p* = 0.1439).

### 3.2. Interhemispheric Downregulation of GAD1 Neurons in the Deprived S1

Changes in fluorescence of GCaMP6s from identified GAD1 neurons in L5 of the deprived S1 were collected in response to stimulation of the CC. Representations of identified L5 GAD1 neurons and their evoked-response activity are demonstrated in Figure 4. For interhemispheric L5 experiments, we imaged 9 slices from denervated rats (*n* = 3) and 15 slices from control rats (*n* = 4). From these slices, we identified 70 GAD1–GCaMP6s positive neurons in denervated rats and 144 in control rats.

Stimulation of CC induced an average fluorescence change of −0.67 ± 1.67% in denervated rats compared to 0.56 ± 2.06% in control rats (Figure 5A; Mann–Whitney Wilcoxon test, *p* < 0.0001). In addition, we found that CC stimulation evoked fewer neural responses in L5 neurons in denervated rats than in control rats. Only 3 out of 70 (4.29%) GAD1 neurons in denervated rats were deemed responsive to stimulation compared to 35 of 144 (24.31%) GAD1 neurons in control rats (Figure 5B; Chi–squared = 12.927, *p* < 0.0003). The average amplitude change of fluorescence between L5 responsive GAD1 neurons in the denervated and control rats was not statistically different (Figure 5C; 1.87 ± 0.83% in denervated rats, 3.02 ± 2.84%; Mann–Whitney Wilcoxon test, *p* = 0.8385).

These results demonstrate that denervation led to an increase in the activity of L2/3 and L5 GAD1 neurons in response to local network activity, while denervation led to a decrease in the activity of L5 GAD1 neurons in response to interhemispheric stimulation of the CC.

## 4. Discussion

Ample research has found that cortical remapping occurs in the S1 following peripheral denervation. This remapping involves both inhibitory interneurons and excitatory neurons. Interneurons receive both excitatory and inhibitory inputs and project locally within the cortical layers [31]. fMRI of the intact and the deprived S1 of denervated rats have shown bilateral increases in both fMRI and single–unit responses following stimulation of the intact limb. The single unit increases were identified as inhibitory interneurons [9]. Li et al. [15] provided additional evidence demonstrating an upregulation in inhibitory interneurons and identified a potential pathway to restore levels of interneuron activity by inhibiting transcallosal communication. Recently, Cywiak et al. [18] demonstrated that excitation of the deprived S1 with non-invasive brain stimulation [32] and magnetogenetics technologies [33] could alleviate pain and improve performance in rats that previously underwent PNI. Moreover, studies show increases in excitatory neurons in the deprived S1 following stimulation, suggesting a shift in the balance of inhibition and excitation [11,13,14,15,17,34,35].

The development and use of transgenic animals for research has been limited to mice due to numerous biological limitations. Transgenic rats are a relatively newer model organism that serves as a better replicate for human disease. The development of genomic modifications in rats has transcended from using cre technology [36] to zinc finger nucleases [37], transcription activator-like effector nuclease (TALEN) [38], and the more revolutionary CRISPR technique [39]. Our research study necessitated the reliable imaging of GAD1 interneuron activity in brain slices in the deprived S1. To do so, we created a strain of transgenic rats that express genetically encoded calcium sensor GCaMP6s in GAD1 neurons. These novel transgenic rats were used to successfully image calcium dynamics of GAD1 neurons in all layers of the somatosensory cortex, with a specific interest in the activity in L2/3 and L5. Through confocal imaging, we identified two pathways, one intrahemispheric and one interhemispheric, that affected the activity of inhibitory interneurons. 

Despite the novelty of our imaging technique, a meticulous approach is required to interpret GCaMP–associated changes. While increases in GCaMP responses are well established to be correlated to increases in neural activity, the cellular basis of decreases in fluorescence may be less clear. A trend seen in our recordings is mild negative deviations of fluorescent changes from the baseline, specifically in denervated rats after transcallosal stimulation. This can be due to: (1) hyperpolarization responses in neurons that have been shown to decrease GCaMP responses [40], (2) small deflections that are a measure of constant calcium flux, (3) negative changes in the fluorescent signal due to a photo–bleaching effect, and/or (4) temporal resolution of GCaMP probe translates to deflections in the baseline. However, it is critical to use the right approach to extract meaningful information from datasets to remove this bias towards negative deflections in imaging [41].

Most of the interneuron projections are local [42,43]. Thereby, these locally connected inhibitory interneurons communicate within layers and are responsible for the mechanisms of intrahemispheric plasticity. In the current study, intrahemispheric stimulation of L2/3 and L5 in the deprived S1 of denervated rats led to increased inhibitory interneuron activity. Several mechanisms could lead to this phenomenon, including long–term depression of excitatory intracortical synapses [44] and potentiation of inhibitory synapses [45].

The CC transmits bilateral sensory signals to the contralateral hemisphere [34,46]. Disruption in interhemispheric connections can cause maladaptive changes, among them the development of phantom limb pain [47]. After unilateral whisker denervation, stimulation of the intact whisker has been shown to strengthen the synaptic connection between the CC and the remote deprived L5 neurons [14,15]. Changes in the functioning of GABAergic receptors in inhibitory interneurons have also been demonstrated post–injury [20,48] as the reduced presence of GABA in the presynaptic terminal post–injury lowers the action potential threshold of the neurons in the targeted region of the deprived S1 [13,14]. In the current study, a decrease in the activity of the inhibitory interneurons was seen in the deprived S1 of denervated rats compared to that of the controls. The strength of excitation–inhibition from the intact to the deprived cortex through the CC is primarily determined by the activity balance and communication between excitatory neurons and inhibitory interneurons across the cortical hemispheres. Injury leads to decreased activity of inhibitory neurons in the deprived S1 and allows for spontaneous activation of excitatory neurons in the remote S1 interhemispheric target [48].

The differences in network activity seen within local connections are opposite of that observed due to remote, interhemispheric differences. A possible mechanism behind this difference could be due to the nature of inhibitory interneurons having a non–homogenous mechanism of plasticity. For example, studies have shown that many populations of GABAergic interneurons fail to undergo the classical NMDA–mediated mechanisms of synaptic plasticity [49]. Also, the vast diversity in interneuron subtypes with 5 different subclasses accounts for innate differences in plasticity mechanisms [50]. Altogether, these studies suggest that both excitatory neurons and inhibitory interneurons are involved in post–injury plasticity. These changes in intercortical and cortical–cortical communication interfere with normal sensory processing and may be the foundation of sensory dysfunctions [19,51,52,53].

### Clinical Translation

Interneuron dysfunction is involved in a variety of neuropathologies, such as schizophrenia [39], epilepsy [54], Alzheimer’s disease [55,56], autism [57], and phantom limb pain [47,58]. Therefore, transgenic rats, such as the novel ones generated for the current study, would be a valuable tool for investigating such pathologies. This is the first time that the activity of inhibitory interneurons was directly visualized via acute brain slice imaging. Through the generation of our transgenic rats, we were able to identify two separate pathways leading to cortical remapping in the deprived S1. Pharmacological approaches [59] and guided neuroplasticity approaches [60] can be further used to specifically target mechanisms driving the changes in the activity of GAD1 neurons.

## Figures and Tables

**Figure 1 biosensors-12-00383-f001:**
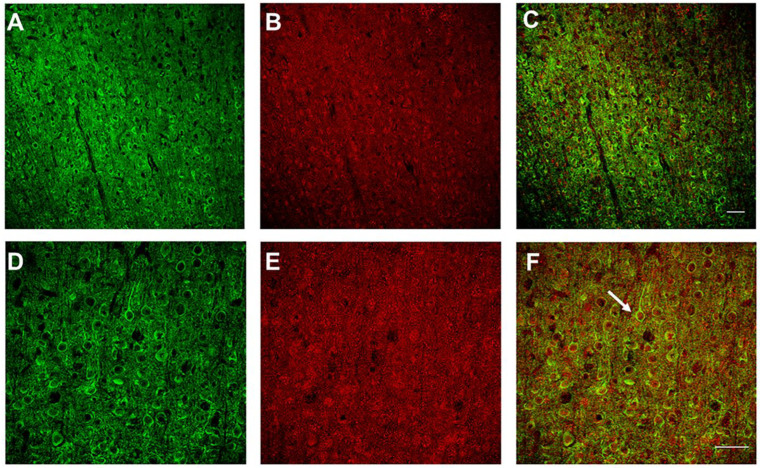
**Immunohistochemistry verification of GAD1-GCaMP6s expression in cortical interneurons.** Double-labeling immunohistochemistry was performed for GCaMP6s (green) and GAD1 (red) in GAD1–GCaMP6s transgenic rats. The top row shows (20×) magnification of a coronal section labeled with (**A**), GCaMP6s antibody (**B**), GAD1 and (**C**), merged (GAD1–GCaMP6s) image. Bottom row shows 40× coronal sections labeled with (**D**), GCaMP6s (green) (**E**), GAD1 (Red), and (**F**), merged GAD1 + GCaMP6s image. The white arrow highlights an interneuron that shows colocalization between GAD1 and GCaMP6s. Scale bars: 50 µm.

**Figure 2 biosensors-12-00383-f002:**
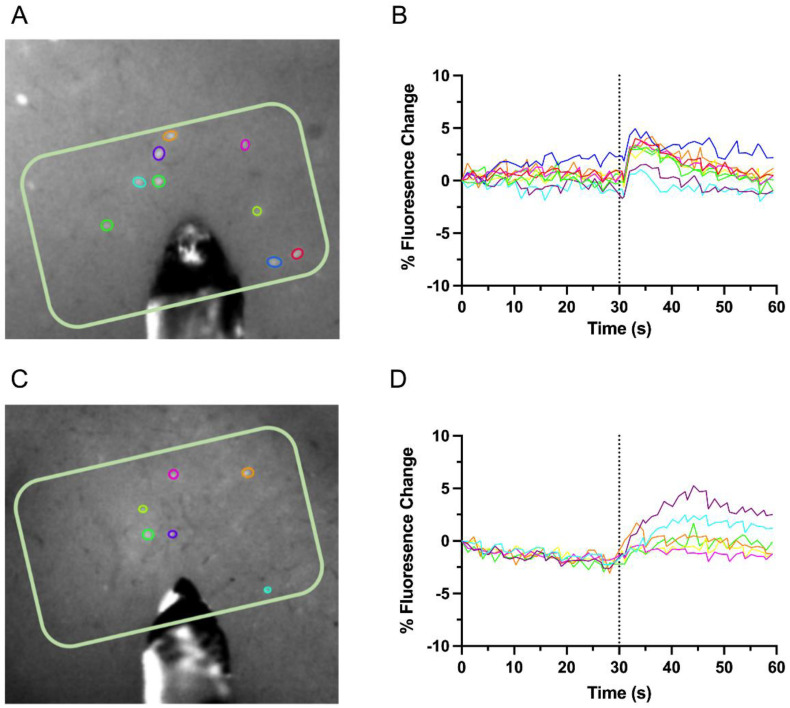
**Intrahemispheric connectivity in L2/3 and L5.** (**A**,**C**), Representative images of L2/3 and L5, respectively, depicting the identified, fluorescing GAD1 neurons (ROIs shown as color coded circle) and their associated; (**B**,**D**), percent fluorescence change over time, with stimulation via bipolar tungsten electrode in the brain slice (pictured in (**A**,**C**)) occurring at 30 s.

**Figure 3 biosensors-12-00383-f003:**
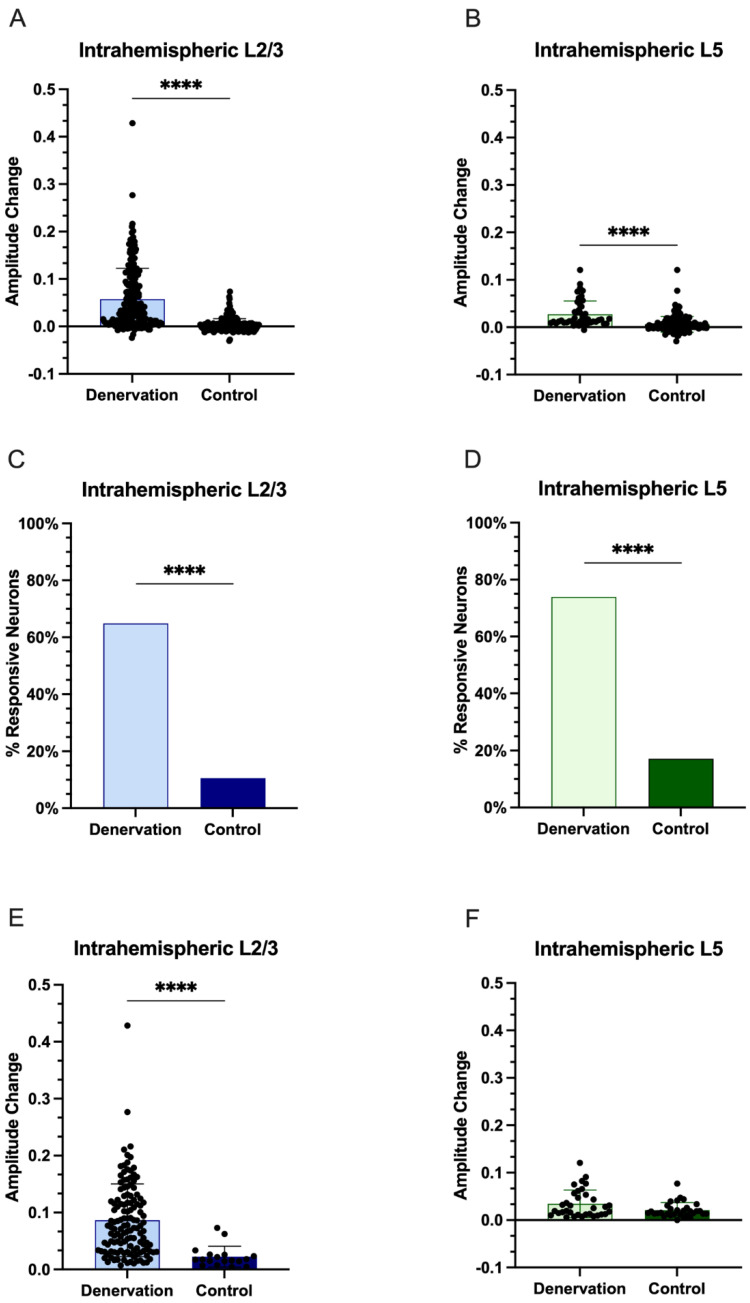
**Intrahemispheric upregulation of GAD1 neuron activity in L2/3 and L5 in the deprived S1 after injury.** (**A**,**B**), fluorescence change (mean + *SD*) of all identified GAD1 neurons after stimulation. (**C**,**D**), number of GAD1 neurons responsive to stimulation, and (**E**,**F**), the average fluorescence change of the responsive GAD1 neurons. (*p* < 0.0001, ****).

**Figure 4 biosensors-12-00383-f004:**
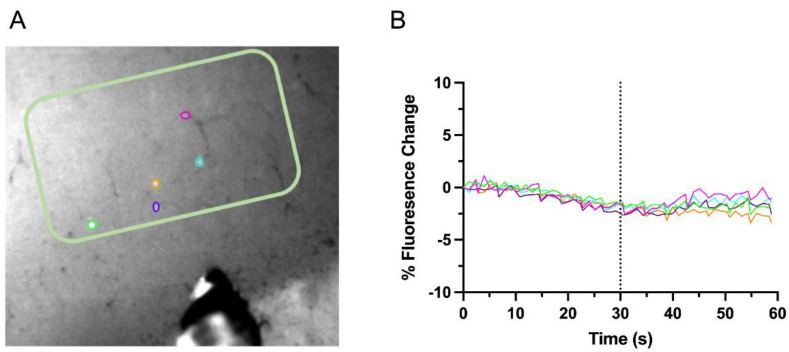
**Interhemispheric connectivity in L5.** (**A**), Representative image depicting the identified GAD1 neurons (ROIs shown as color coded circle). (**B**), Percent fluorescence change over time, with stimulation via bipolar tungsten electrode in the brain slice (pictured in (**A**)) occurring at 30 s.

**Figure 5 biosensors-12-00383-f005:**
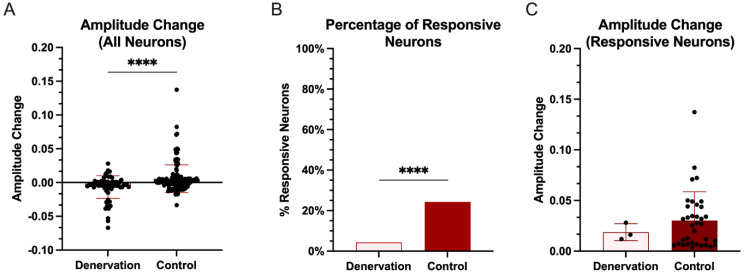
**Interhemispheric downregulation of GAD1 neuron activity in L5 in the deprived S1 after injury.** (**A**), Fluorescence change (mean + SD) of all identified GAD1 neurons after stimulation, (**B**), number of GAD1 neurons responsive to stimulation, and (**C**), the average fluorescence change of the responsive GAD1 neurons. (*p* < 0.0001, ****).

## Data Availability

Raw and analyzed data will be shared upon request. We would expect that upon completing their independent data analysis, researchers would cite our published work and/or provide co-authorship as appropriate.

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
