# Peer review of "Peripheral Nerve Injury Induces Changes in the Activity of Inhibitory Interneurons as Visualized in Transgenic GAD1-GCaMP6s Rats"

_biosensors, 2022, doi:10.3390/bios12060383_

Round 1

Reviewer 1 Report

See attachment

Reviewer 2 Report

Abstract: The authors should go ahead and define 'local' in regards to intra hemispheric stimulation. We're left to assume that this is in the somatosensory cortex until much later.

2.4 Confocal Imaging: It is unclear why the authors reported the temporal resolution of fixed tissue imaging, or what this value means. The more relevant parameter here would be pixel dwell time but, even then, it would only be useful in combination with settings (e.g., 512x512).

Two weeks is cutting it very close in regards to acute/chronic injury development. Were any behavioral experiments conducted to determine which animals may be suffering from chronic pain phenomena at the two week time point (e.g., autotomy scores)? If you are studying plasticity changes in the brain that occur due to chronic pain, we need to know that chronic pain is presented.

Figure 2: Since this is a tissue slice, it would be useful to see a trace for an ROI not identified as an inhibitory interneuron.

Responsive threshold: Is the Baseline fluorscense detrended prior to the calculation of 2SD?

Figure 5: Something is drastically off between the numbers provided and the graph in Figure 5A. There is at least an order of magnitude difference between the mean values (reported versus plotted) and the reported standard deviations swamp the means by nearly an order of magnitude - making the reported p-value impossible. The first point may simply be a matter of y-axis units (ratio versus %), but the second point is not.

Introduction and conclusion: There should be some discussion of prior work regarding genetically modified animal models targeting GAD1. Without appropriate context, the claim of novelty is too bold. There are a number of published GAD1 transgenic models, both mice and rats, as well as a few mouse models that incorporate GCaMP. The novelty appears to be in the method used to create the strain and the specific GCaMP used.

The stated goal is not well aligned with the study itself. As indicated in the concluding introductory paragraph, the presented work provides evidence that activity is altered - not in what way or due to any specific mechanisms. The innovation here is presumably in the model organism, not the insight into interneuron activity following injury.

Round 2

Reviewer 2 Report

The authors were responsive on several key points, especially regarding contextualization of their model. However, the authors were not at all responsive to questions raised about their data; especially in regards to previous points 5 and 6.

Without detrending the data, you are superficially increasing the 2SD threshold for responsiveness – the average of time-dependent decreasing (bleaching) data is larger than it would be under stable conditions. Moreover, there is no interpretation given as to why stimulation of denervated tissue should result in reduced (negative mean and, presumably, median) fluorescence. Is there calcium efflux? That is not the case. It is, instead, that the data is trending in the negative direction. Below is a crude illustration of the point from Figures 4 and 2. Your innervated control group will still have larger increases, but the negative values are nonsensical – or, if they are not, need to be explained.

*Figure in attached word document

Along those lines, the data does not seem to follow a normal distribution. What tests were run to accept or reject normality before proceeding with the student t-test?

Figure 2 does not seem to be referenced at all in the text.
